# The Comparative Evaluation of the Fujifilm Wako β-Glucan Assay and Fungitell Assay for Diagnosing Invasive Fungal Disease

**DOI:** 10.3390/jof9010006

**Published:** 2022-12-20

**Authors:** Shreya Singh, Rimjhim Kanaujia, Sourav Agnihotri, Harsimran Kaur, Arunaloke Chakrabarti, Shivaprakash M. Rudramurthy

**Affiliations:** Department of Microbiology, Postgraduate Institute of Medical Education and Research, Chandigarh 160012, India

**Keywords:** 1,3-β-d-glucan, invasive fungal disease, invasive aspergillosis, invasive candidiasis, fungal biomarker, Fujifilm Wako assay, Fungitell assay

## Abstract

Serum 1,3-β-d-glucan(BDG) is a broad fungal biomarker for invasive fungal disease (IFD). More data is still required to support the Fujifilm Wako assay as a valuable alternative to the widely used Fungitell assay. We included archived serum samples from 157 individuals (97 cases; 33-IA, 64-IC, and 60 controls) for the comparative performance evaluation of the Fungitell assay and the Fujifilm Wako assay for IFD diagnosis. The BDG value was significantly higher in patients with IFD vs. controls (70.79 pg/mL vs. 3.03 pg/mL, *p*: 0.0002). An area under the curve (AUC) for the IFD, IC, and IA diagnosis was 0.895, 0.910, and 0.866, respectively, for the Fujifilm Wako assay. Based on the highest Youden’s index (0.667), a cutoff of 5 pg/mL was selected as the optimum for the Fujifilm Wako assay with good sensitivity (79.4%), specificity (88.3%) and agreement (84.7%, Cohen’s k:0.691) with the Fungitell assay. The mean run-time of the Fujifilm Wako assay was 70.12 min, and real-time observation allowed earlier time to result in Fujifilm Wako vs. Fungitell assay (37 vs. 120 min, *p:* < 0.0001). Thus, our findings support the diagnostic value of the Fujifilm Wako assay for the diagnosis of IFD. However, there is still a need to validate diagnostic protocols to optimize their use in multi-centre studies with different patient groups.

## 1. Introduction

The prompt diagnosis of fungal infections is imperative to ensure good patient outcomes. In view of the limitation of traditional diagnostic methods such as low sensitivity, long turnaround time and the need for invasive sampling, the use of biomarker-based diagnostic approaches has gained much importance [1]. Among the available fungal biomarkers, the detection of serum 1,3-β-d-glucan (BDG) has been reported to help in the diagnosis of almost all invasive fungal diseases (IFD), except for mucormycosis and cryptococcosis [1,2]. Various kits are commercially available, including the Fungitell assay (Associates of Cape Cod, East Falmouth, MA, USA) and Fujifilm Wako β-glucan test (FUJIFILM Wako Pure Chemical Corporation, Osaka, Japan). Both these tests rely on the basic principle of factor G activation in response to BDG in the Limulus (horseshoe crab) coagulation cascade pathway. Factor G, a zymogen enzyme, activates the pro-clotting pathway resulting in the cleavage of an artificial substrate used in both assays. The detection and estimation of BDG can be performed by either assay based on the colorimetric (Fungitell Assay) or turbidimetric (Fujifilm Wako assay) principle. The proposed cutoff of 80 pg/mL (Fungitell Assay) and 11 pg/mL (Fujifilm Wako assay) has shown that the Fujifilm Wako assay is specific, whereas the Fungitell Assay is more sensitive for the diagnosis of Pneumocystis pneumonia (PCP). The sensitivity of the Fujifilm Wako assay was improved by lowering the cutoff to as low as 3.616 pg/mL for diagnosis of IFD, including PCP and invasive candidiasis (IC) [3,4]. In a recent study, using an optimized cutoff of 7.0 pg/mL, the sensitivity and specificity of the Fujifilm Wako assay were 80.0% and 97.3% for diagnosing invasive aspergillosis (IA) and 98.7% and 97.3% for IC, respectively [5]. However, more data is needed to support using the Fujifilm Wako assay as a promising alternative to the Fungitell assay, especially for patients with IC and IA.

Therefore, we planned to compare the performance of the Fujifilm Wako assay with the Fungitell assay in well-characterized groups of patients with IA and IC in reference to appropriate control patients. We also attempted to define the optimal cutoff values for the Fujifilm Wako assay to exclude IFD reliably.

## 2. Material and Methods

We used archived patients’ serum samples, which were collected and stored as part of routine clinical care at the Postgraduate Institute of Medical Education and Research (PGIMER). All the patients with a first serum BDG sample collected for routine mycological testing were initially considered eligible.

Case selection: We included all the adult patients classified as IFD (proven or probable, IC or IA). Proven cases were defined based on the presence of fungal hyphae or yeast on histopathology, cytopathology, or direct microscopic examination of sterile specimens accompanied by evidence of tissue damage. We also considered the recovery of yeast or mold from sterile samples consistent with an infectious disease process as proven IFD. For the diagnosis of probable IFD, we used the revised EORTC/MSGERC definitions among immunocompromised patients [6], Bulpa criteria for diagnosing IA in patients with COPD [7], modified AspICU Blot criteria for critically ill patients admitted to ICU’s [8], and the Modified AspICU Blot criteria with the inclusion of COVID-19 as host entry criteria were used for diagnosing Coronavirus associated pulmonary aspergillosis (CAPA) [9].

Control selection: Serum samples from patients with risk factors for IFD who did not meet the criteria for proven or probable disease (i.e., who had no evidence of IFD) were included as controls.

All included samples from both cases and controls were stored within two days after obtaining the evidence or lack of proof of defining IFD. The samples were stored at −80 °C until the testing, which was within two weeks of their collection. A blinded technician performed both tests simultaneously with no information about any patient’s IFD classification at the time of testing.

### 2.1. Serum BDG Measurement

The frozen serum samples were thawed, brought to room temperature, and vortexed briefly before testing. Both assays were performed in accordance with the manufacturer’s instructions. A positivity threshold of 80 pg/mL was used for the Fungitell assay as directed by the manufacturer. For the Wako BDG, 100 μL of serum sample was added to 900 µL of pretreatment solution. After incubation at 37 °C for 10 min on a Thermo Station (Fujifilm Wako), the sample was cooled on the Cooling Station (Fujifilm Wako). Two hundred microliters of the pretreated sample was then added to the Limulus amebocyte lysate (LAL) reagents. Kinetic turbidimetric results were measured for 90 min at 37 °C using a machine for BDG measurements with a single extension module (Fujifilm Wako Pure Chemical Corporation, Osaka, Japan). BDG concentration was calculated by comparing the gelation time with the manufacturer’s calibration curve provided with each lot. The optimum positivity threshold or cutoff was determined on a subsequent analysis for the Fujifilm Wako assay. Fungitell is a batch assay, and Fujifilm Wako Beta-D-Glucan is a single-test assay

### 2.2. Data Collection and Statistical Analysis

Statistical analysis was performed using GraphPad Prism version 8 and SPSS Statistical Software version 23. Patients’ demographics and BDG data were expressed as appropriate proportions/mean/median. The sensitivity and specificity were determined by constructing 2 × 2 tables using IFD (IA or IC) patients as true cases and non-IFD patients as controls. Receiver operating characteristic (ROC) curves were generated for both assays. We used the highest Youden index to derive the optimal BDG cutoff for the Fujifilm Wako assay. Agreement between BDG assays values was determined by the Bland–Altman correlation plot, whereas Cohen’s kappa statistics were used for the strength of agreement.

## 3. Results

A total of 157 individuals, including 97 patients with IFD (33—IA and 64—IC) and 60 non-IFD controls, were included in the study. Among the patients with IFD, 12 had proven IFD (3 IA and 9 IC), while the remaining cases were of probable IFD. Since the number of proven cases was few, this subgroup was not analyzed further. Overall, the mean age of the participants was 40.8 ± 16.4 years, and there were 63% males. Among the cases with IFD and non-IFD controls, the mean ages (in years) were 36.7 ± 12.8 and 43.5 ± 17.8, with 61.8% and 65% males, respectively. The mean BDG levels groups using the Fujifilm Wako assay in various patient groups are depicted in Figure 1. A significantly higher BDG value was noted in patients with IFD vs. controls (70.79 vs. 3.03, *p*-value: 0.0002), IA vs. controls (112.3 vs. 3.034, *p*-value: <0.0001), and IC vs. controls (49.4 vs. 3.034, *p*-value: 0.0009).

The comparative values of BDG measured using both kits are depicted in Figure 2A. The Bland–Altman curve analysis of the BDG levels obtained by both methods revealed a bias of 137.43 (SD 242.7, limits of agreement: −513.1 to 438.3). The Bland–Altman scatterplot shown in Figure 2B compares the measurements of the BDG using both kits, with the X-axis depicting the mean/average of both measurements and the Y-axis representing the difference between the two measurements. The scatterplot was evaluated according to the scatter dispersion; less scattering indicates a good agreement, as the individual points lie relatively close to the line representing mean bias. It is recommended that 95% of the data points lie within ±1.96 SD of the mean difference, i.e., the limits of agreement. The chart indicates that while the actual estimates of BDG were higher in Fungitell for most samples, all except three values lie between the recommended levels of agreement.

Based on Youden’s index, a 5 pg/mL cutoff was selected as the optimal cutoff for the Fujifilm Wako assay, with a sensitivity and specificity of 79.4% and 88.3%, respectively (Table 1). At this cutoff, the % agreement between the Fujifilm Wako and Fungitell assay was 84.7%, with a Cohen’s k score of 0.691.

The ROC curve with the area under the curve (AUC) for the Fungitell and Fujifilm Wako assay is shown in Figure 3. A good performance with an AUC of 0.990 for the Fungitell and 0.895 for the Fujifilm Wako assay was seen.

### 3.1. Assay Time Assessment

The mean run-time of the Fujifilm Wako assay was 70.12 min. The time to result for BDG estimation using the Fujifilm Wako assay is depicted in Figure 4, indicating that the BDG values were inversely proportional to the time to result. So, the real-time observation of BDG values in the Fujifilm Wako assay allowed earlier detection of BDG positive samples with a mean time to result of 37 min (median 34.8 min, range 8–90) compared to 120 min for the Fungitell assay, respectively; *p*-value < 0.0001).

The mean time to result in the BDG negative samples was 87 min using the Fujifilm Wako assay (median 90 min, range 53–90 min) and 120 min with the Fungitell assay (*p*-value < 0.0001). On including the initial 15-min pre-incubation, the time to positive results was still significantly faster on using the Fujifilm Wako assay vs. Fungitell assay (57.26 min vs. 120 min, respectively for positive samples; *p* < 0.0001 and 93.15 vs. 120 min, respectively for negative samples; *p* < 0.0001).

### 3.2. Subgroup Analysis

For the diagnosis of IC (n = 64), the Fujifilm Wako assay showed an AUC of 0.910, while it was 0.998 for the Fungitell assay (Figure 5A). In these patients, the sensitivity and specificity of the Fujifilm Wako assay were 59.4% and 100% at the manufacturer’s cutoff of 11 pg/mL and 81.3% and 88.3% at the cutoff of 5 pg/mL. The mean time to result (excluding pre-incubation time) was 45.9 min for positive samples and 84.2 min for negative samples.

Among patients with IA (n = 33), the Fujifilm Wako assay showed an AUC of 0.866, while it was 0.974 for the Fungitell assay (Figure 5B). In these cases, the sensitivity and specificity of the Fujifilm Wako assay were 57.6% and 100% at the manufacturer’s cutoff of 11 pg/mL and 75.8% and 88.3% at the determined cutoff of 5 pg/mL. The mean time to result (excluding pre-incubation time) was 44.5 min for positive samples and 84.2 min for negative samples.

## 4. Discussion

In this study, we performed a retrospective performance evaluation of the turbidimetric Fujifilm Wako assay to diagnose IFD (including IA and IC). We observed substantial agreement (84.7%) between the Fujifilm Wako and Fungitell assay, with a good AUC of 0.895 for the Fujifilm Wako assay. A 5 pg/mL cutoff was optimum for the diagnosis of IFD using the Fujifilm Wako assay with a sensitivity of 79.4% and specificity of 88.3%. Previously, a study by Cento et al. revealed that the single quantification of serum BDG using the Fujifilm Wako assay had a sensitivity of 56%, specificity of 87%, NPV of 85%, and PPV of 59% at the cutoff of 11 pg/mL in critically ill ICU and non-ICU patients [10]. Lowering the cutoff to 7.7 pg/mL, the sensitivity, specificity, NPV, and PPV were 56%, 87%, 85%, and 59%, respectively, similar to our findings [10]. Lower cutoffs for the Fujifilm Wako assay have been proposed previously. The cutoff ranging from 3.6 pg/mL to 11 pg/mL demonstrated satisfactory sensitivity, higher specificity, and better technical flexibility than the Fungitell assay [3,4,11]. In specific IFDs, the Fujifilm Wako assay (cutoff 11 pg/mL) demonstrated a sensitivity and specificity of 60% and 99.5%, for diagnosing IA and 91% and 99.5%, for IC, respectively [5]. Further optimization of the cutoff to 7 pg/mL, improved the sensitivity and specificity to 80.0% and 97.3% for IA diagnosis and 98.7% and 97.3% for IC diagnosis, again suggesting a need for a lowering the cutoff.

In a recent evaluation of five new BDG measuring kits, including Fungitec G test ES (Nissui Pharmaceutical Co., Ltd., Tokyo, Japan), Fungitell β-D-glucan assay (Associates of Cape Cod, MA, USA), Fungitec G test MKII (Nissui Pharmaceutical Co.), β-Glucan test Wako (Wako, FUJIFILM, Pure Chemical Corporation, Osaka, Japan), and Wako-Eu kit (Wako), the Fujifilm Wako BDG assay demonstrated a higher specificity and PPV valuewhile the Fungitell assay had more false positives [12]. Establishing a lower cutoff level can increase false positive test results, thus requiring cautious interpretation. Previously, authors have demonstrated concordance between both the Fujifilm Wako and Fungitell assays in patients with IFD (95.6% agreement; Cohen’s kappa: 0.91), IA (95.1% agreement; Cohen’s kappa: 0.83), and IC (97.3% agreement; Cohen’s kappa: 0.93) which is similar to our findings [5].

The basis of BDG measurement in both the Fungitell assay and the Fujifilm Wako assay is the Limulus amebocyte lysate coagulation cascade pathway. However, there are differences in the testing protocol and final detection chemistries, which results in different estimates of BDG from the same sample. We observed that the values of BDG estimated by Fungitell were higher than those detected by the Fujifilm Wako assay, and therefore, the optimum cutoff values need to be standardized in specific patient groups using both these assays separately. We noted that the majority of the estimates of BDG using both assays were within the ±1.96 SD of the mean difference on Bland–Altman analysis which suggests good agreement between both assays despite the difference in actual BDG values.

We also compared the layout and workflow of both assays, and some of the good technical characteristics of the Fujifilm Wako assay are, firstly, a random-access approach that allows parallel runs of up to 16 serum/plasma samples even on sequential loading onto the instrument. Secondly, a predetermined internal calibration curve which obviates the requirement for single-run calibrations and running standards with each run. Lastly, a shorter run time (mean: 70.12 min and maximum: 120 min) allows the laboratory to provide rapid results, which helps to make timely clinical decisions.

The Fungitell assay is the only commercially available BDG kit in the USA and Europe, while Fujifilm Wako has been recently made available in European countries. However, no studies evaluating the Fujifilm Wako assay are available from India. Our study is the first to provide valuable information regarding the potential use of this assay in our region. In light of our findings, the Fujifilm Wako assay seems to hold promise for the diagnosis of IFD, and our results indicate its diagnostic value in clinical settings. However, there is still a need to establish and validate diagnostic protocols to optimize the Fujifilm Wako assay’s diagnostic cutoff using multi-centre studies in different patient groups.

We acknowledge certain limitations of the study. First, the study was retrospective and was performed on stored serum samples. However, we tried to address this by ensuring that both tests were performed simultaneously and within two weeks of sample storage. Secondly, due to the lack of invasive diagnostics and autopsy findings, some cases of IFD may have been falsely categorized as non-IFD, contributing to false positive results in the study. Thirdly, the majority of the cases of IFD were probable and not proven cases. This is because archived serum samples were used and patients with proven IFD are infrequently subjected to biomarker-based testing, which is usually done when conventional tests are not conclusive. Additionally, the need for invasive sampling often required to establish a diagnosis of proven IFD limits the availability of proven IFD cases. Lastly, only one serum sample was tested in all patients (i.e., the first serum sample even if more samples were collected) and repeat testing could have influenced test sensitivity. Nonetheless, with this study, we provide valuable data about the role of serum BDG in the diagnosis of IFDs employing a comparative assessment of both the Fujifilm Wako and Fungitell assay in terms of sensitivity, specificity, and technical flexibility in a large number of samples which can be a guide for infectious disease practice.

## 5. Conclusions

The overall performance of the Fujifilm Wako assay seems satisfactory, with a sensitivity of nearly 80%, specificity of 88.3%, and AUC of 0.895 at a BDG value cutoff of 5 pg/mL with a good (84.7%) agreement between the Fujifilm Wako and Fungitell assay. The Fujifilm Wako assay also had a significantly shorter run time compared to Fungitell and the real-time estimation of BDG allows earlier detection of positive samples. As per our assessment, the Fujifilm Wako assay seems to be a valuable alternative to the Fungitell assay. Multi-centre coordination, test validation in specific patient groups, and repeated sampling could further add to the clinical applicability of this assay in future studies.

## Figures and Tables

**Figure 1 jof-09-00006-f001:**
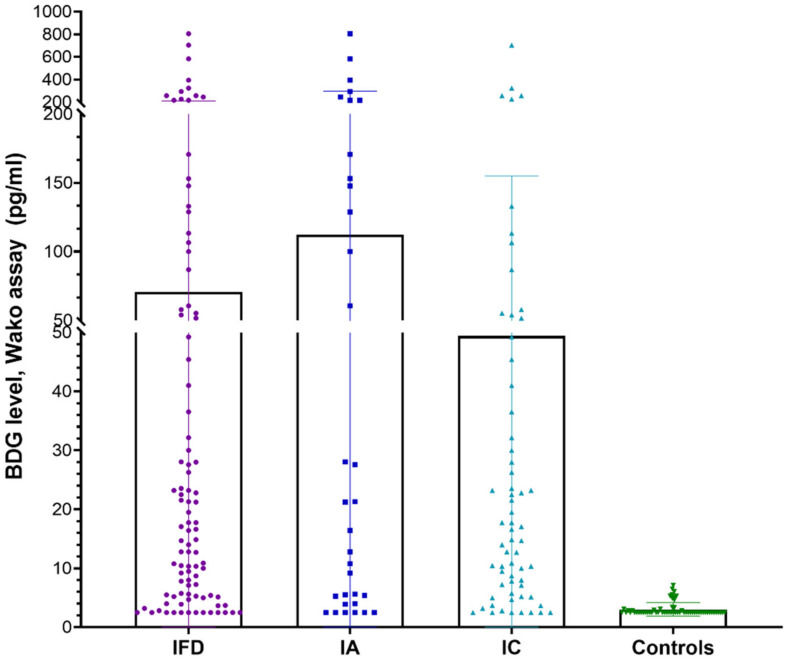
Comparative BDG levels were observed in various patient groups using the Fujifilm Wako BDG estimation assay.

**Figure 2 jof-09-00006-f002:**
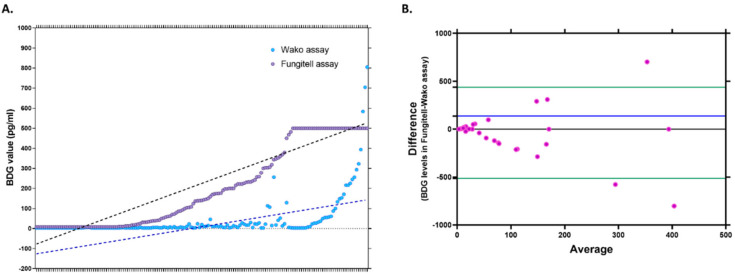
(**A**). Values of BDG levels of individual samples estimated using the Fungitell and Fujifilm Wako assay. Here the dotted lines represent the BDG value data trends in respective assays (**B**). Bland–Altman scatterplot showing the dispersion of individual sample estimates of BDG measured using the Fungitell and Fujifilm Wako assay. Here, the blue line indicates the mean bias, green lines: upper line indicates theupper limit of agreement, and the lower line indicates thelower limit of agreement.

**Figure 3 jof-09-00006-f003:**
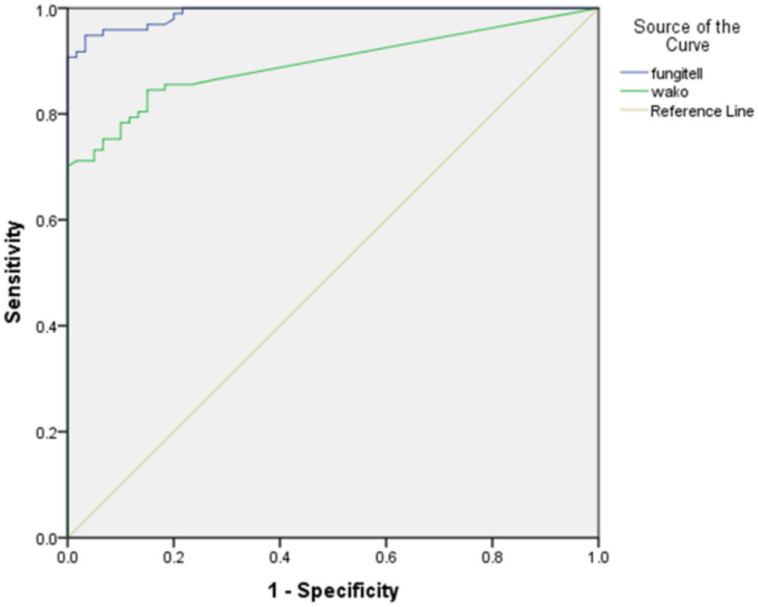
Receiver operating characteristics (ROC) curve showing the sensitivity and false positive rates (1–specificity) in detecting IFD (proven/probable) using the Fungitell BDG assay (blue) and Fujifilm Wako assay (green).

**Figure 4 jof-09-00006-f004:**
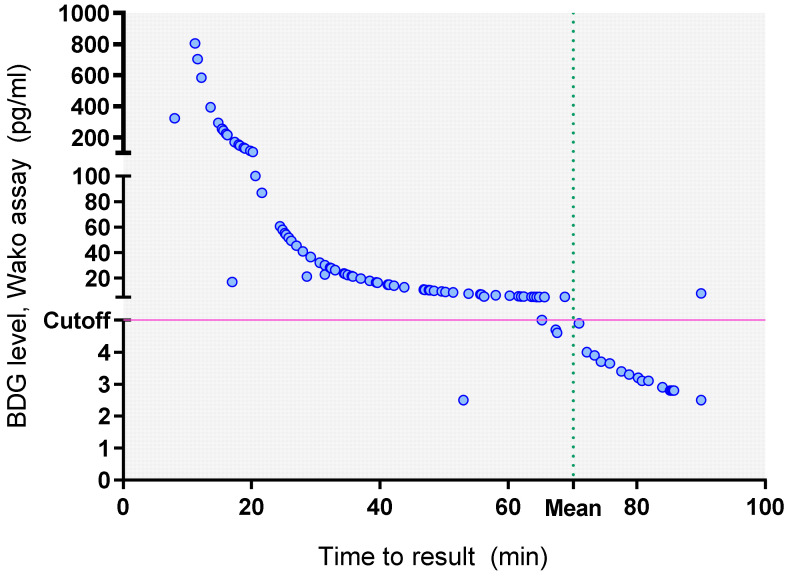
Individual BDG levels estimated using the Fujifilm Wako assay and their corresponding time to result (minutes).

**Figure 5 jof-09-00006-f005:**
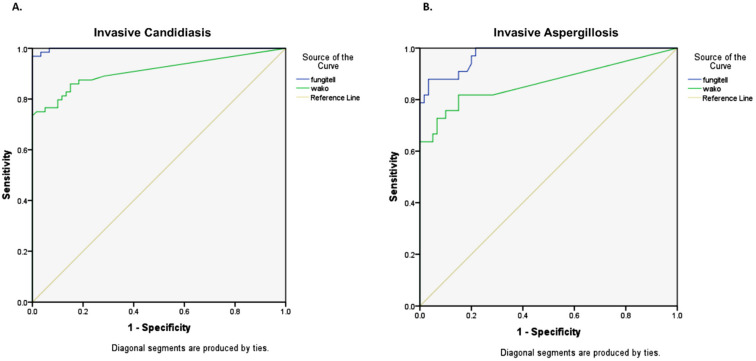
Receiver operating characteristics (ROC) curve showing the sensitivity and false positive rates (1–specificity) in detecting IFI (proven/ probable) using the Fungitell BDG assay (blue) and Fujifilm Wako assay (green) in patients with (**A**) Invasive candidiasis and (**B**) Invasive aspergillosis.

**Table 1 jof-09-00006-t001:** The diagnostic performance characteristics of the Fungitell assay at the manufacturer’s cut-off and Fujifilm Wako assay at various cut-off values for diagnosis of invasive fungal disease.

BDG Test Assay	Sensitivity	Specificity	Positive Likelihood Ratio	Negative Likelihood Ratio	Positive Predictive Value	Negative Predictive Value	Youden’s Index
Fungitell assay; (Kit cut-off 80 pg/mL)	91.75%; (84.39% to 96.37%)	98.33%; (91.06% to 99.96%)	55.05; (7.88 to 384.82)	0.08; (0.04 to 0.16)	98.89%; (92.72% to 99.84%)	88.06%; (79.14% to 93.48%)	0.901
Fujifilm Wako assay; (Kit cut-off 11 pg/mL)	59.79%; (49.35% to 69.63%)	100.00%; (94.04% to 100.00%)	-	0.40; (0.32 to 0.51)	100.00%	60.61%; (54.69% to 66.23%)	0.598
Fujifilm Wako assay; (cut-off 5 pg/mL)	79.38%; (69.97% to 86.93%)	88.33%; (77.43% to 95.18%)	6.80; (3.37 to 13.75)	0.23; (0.16 to 0.35)	91.67%; (84.48% to 95.70%)	72.60%; (63.95% to 79.83%)	0.677
Fujifilm Wako assay; (cut-off 4 pg/mL)	81.44%; (72.27% to 88.62%)	85.00%; (73.43% to 92.90%)	5.43; (2.95 to 9.99)	0.22; (0.14 to 0.34)	89.77%; (82.67% to 94.17%	73.91%; (64.82% to 81.33%)	0.664

## Data Availability

The data will be made available on reasonable request.

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
