# Peer review of "The Comparative Evaluation of the Fujifilm Wako β-Glucan Assay and Fungitell Assay for Diagnosing Invasive Fungal Disease"

_jof, 2022, doi:10.3390/jof9010006_

Round 1

Reviewer 1 Report

The paper by Singh describes a comparative study between Fujifilm Wako β-glucan assay and Fungitell assay for diagnosing invasive fungal disease. The paper deals with an interesting and timely topic such as accurate invasive aspergillosis and invasive candidiasis diagnosis. However, some major points should be addressed to support the conclusions of this study.

Major concerns:

- It has been shown that b-1,3-glucan from the diet (cereals, bread/beer yeast, other fungi and plants consumed) is absorbed in the intestine through Peyer's patches into the relevant lymphatic vessel and ultimately into the bloodstream.

It would be very interesting to see if this beta-glucan from the diet is positive in these tests as it could destroy the usefulness of both diagnostic kits, which have always been in question.

o             https://www.ncbi.nlm.nih.gov/pmc/articles/PMC3236515/

o             https://www.preprints.org/manuscript/202012.0250/v1/download

- Authors say that any patient sample can be eligible from the first routine analysis of the patient. How did they choose them? Was it random or did they take the ones with the highest amount of beta-glucan in Fungitell?

- Line 65: Wouldn't it be better to use healthy patients as controls? Platelia or Fungitell tests are routinely done on patients at risk. In fact, when they are positive in one of them, IFI is suspected. Perhaps it is not a good idea to rule out Fungitell negatives by default. Healthy patients should be the control, and patients at risk but with no evidence of IFI should be ruled out, predicted by the test but not used as a control.

- The results by subgroups as specified by the manufacturer are not very good... The specificity is 100%, but the sensitivity... All the ones it determines are good, but it does not detect half of them.

- It would be interesting if the authors, while keeping their groups as the sum of possible and proven, could separate their groups of IFIs into possible and proven to see how well it works in the proven ones.

Minor concerns:

- Line 15: AUC should not be shown only in abbreviated form.

- Figure 1: It would be nice to see the 0 to 200 pg/mL scale better, otherwise, it is a graph to show the 3 outsiders.

Author Response

Please see the attachmentPlease see the attachment.

Reviewer 2 Report

In this study, the authors evaluate the efficacy of Fujifilm Wako assay for the detection of fungal pathogens in patient samples. The retrospective study is well-designed, and this reviewer commends the efforts put forth by the authors in the planning and execution of the study. The study is also timely and will be beneficial for the scientific community and especially for hospital settings and diagnostic laboratories. However, there are a few minor concerns that should be addressed for clarity (listed below). There are several grammatical errors in the entire manuscript (some of which are mentioned in the list below) and it is suggested the authors perform thorough proofreading during revision.

 General:

1.       References should be placed before the period (.) at the end of the sentence and not after the period.

2.       Temperature should be written as 37oC (degrees C) not just 37C

3.       Introduction is too brief and should be expanded. Rather than directly focusing on the two assays compared in this study, a brief report on the existing technologies currently available for pathogen detection (hospital and diagnostic labs), and their limitations/advantages will be helpful.

4.       Overall, there is only limited reference to previously published reports.

5.       Lines 35-36: Please re-word the sentence. It is incomplete.

6.       Line 39: should read as “The sensitivity of the Fujifilm Wako assay was improved..”

7.       Lines 70-72: How many times was each sample tested? Was it just one time by a single technician as reported?

8.       Line 79: Should be “cooled” not “colled”

9.       Line 80: should be “sample was”

10.   Figure 1: This figure gives an assumption that the BDG level in true positives (IA and IC) is zero for several patients. A split y-axis can better depict the results within the 0-100 or 200 range. This can easily be done in GraphPad.

11.   Figure 2: What was the rationale for not showing the time to result for Fungitell BDG assay

12.   Figure 5: While estimating the AUC, what was the cutoff employed for Fungitell assay? Was it the manufacturer cutoff or a value was determined for this assay too just like it was lowered to 5 pg/ml instead of 11 pg/ml for Wako assay? Based on the data reported in this figure, the specificity and sensitivity seem to be compromised in Wako assay when compared to Fungitell assay. This would raise serious concerns regarding the feasibility and applicability of Wako assay diagnostics as stated in the discussion (Lines 168-171). Further, a lower cutoff value increases the false positive rate which can be misleading for treatment.

13.   Line 185: Please mention the names of all four BDG assay kits used in this study.

14.   Lines 193-195: Incomplete sentence. Should be reworded.

15.   Lines 193-194 and 194-195 are contradicting each other. If it is the same principle, why should there be differences in detection? Rephrase the sentence.

16.   Line 203: should read as “…and some of the good technical…”

17.   Table 1 should be mentioned in the text.

Round 2

Reviewer 1 Report

Thank you very much for your answers.